# Inhibition of lung tumorigenesis by a small molecule CA170 targeting the immune checkpoint protein VISTA

Jing Pan [1,2,3,9], Yao Chen[4,5,9], Qi Zhang [1,2,3,9], Achia Khatun[4,5], Katie Palen[6], Gang Xin[4,5], Li Wang[7], Chuanjia Yang[1,2], Bryon D. Johnson[6], Charles R. Myers[1,2], Shizuko Sei [8], Robert H. Shoemaker[8], Ronald A. Lubet[8], Yian Wang[1,2,3], Weiguo Cui[4,5✉] & Ming You [1,2,3✉]

Expressed on cells of the myeloid and lymphoid lineages, V-domain Ig Suppressor of T cell Activation (VISTA) is an emerging target for cancer immunotherapy. Blocking VISTA activates both innate and adaptive immunity to eradicate tumors in mice. Using a tripeptide small molecule antagonist of VISTA CA170, we found that it exhibited potent anticancer efficacy on carcinogen-induced mouse lung tumorigenesis. Remarkably, lung tumor development was almost completely suppressed when CA170 was combined with an MHCII-directed KRAS peptide vaccine. Flow cytometry and single-cell RNA sequencing (scRNA-seq) revealed that CA170 increased CD8$^+$ T cell infiltration and enhanced their effector functions by decreasing the tumor infiltration of myeloid-derived suppressor cells (MDSCs) and Regulatory T (Treg) cells, while the Kras vaccine primarily induced expansion of CD4$^+$ effector T cells. VISTA antagonism by CA170 revealed strong efficacy against lung tumorigenesis with broad immunoregulatory functions that influence effector, memory and regulatory T cells, and drives an adaptive T cell tumor-specific immune response that enhances the efficacy of the KRAS vaccine.

[1] Center for Disease Prevention Research, Medical College of Wisconsin, Milwaukee, WI, USA. [2] Department of Pharmacology & Toxicology, Medical College of Wisconsin, Milwaukee, WI, USA. [3] Center for Cancer Prevention, Houston Methodist Cancer Center, Houston Methodist Research Institute, Houston, TX, USA. [4] Versiti Blood Research Institute, Milwaukee, WI, USA. [5] Department of Microbiology & Immunology, Medical College of Wisconsin, Milwaukee, WI, USA. [6] Department of Medicine, Medical College of Wisconsin, Milwaukee, WI, USA. [7] Department of Translational Hematology and Oncology Research, Cleveland Clinic Foundation, Cleveland, OH, USA. [8] Chemopreventive Agent Development Research Group, Division of Cancer Prevention, National Cancer Institute, Bethesda, MD, USA. [9] These authors contributed equally: Jing Pan, Yao Chen, Qi Zhang. ✉email: wecui@mcw.edu; myou@houstonmethodist.org

Immune checkpoints limit the strength and duration of immune responses and are important to maintain balance between tolerance and autoimmunity. During T cell activation, inhibitory receptors such as cytotoxic T-lymphocyte-associated protein 4 (CTLA-4), Programmed cell death protein 1 (PD-1), Lymphocyte-activation gene 3 (Lag-3), T cell immunoglobulin and mucin domain-containing protein 3 (Tim-3), T cell immunoreceptor with Ig and ITIM domains (TIGIT) and V-domain Ig suppressor of T cell activation (VISTA) are induced to limit overstimulation of the immune system after antigen exposure[1]. The ligands for these inhibitory receptors are upregulated in many types of cancer cells, antigen-presenting cells (APCs), and other immune cells, and the binding of these ligands to their receptors usually results in reduced T cell proliferation and activation. These immune checkpoints are functionally nonredundant since they act at different locations and times and on different immune cell types during immune responses[2,3].

VISTA is an immunotherapy target for both innate and adaptive immunity and has limited homology to other B7 family members. VISTA is most highly expressed on myeloid and granulocytic cells[4]. VISTA controls T cell activation through nonredundant functions distinct from the PD-1/PD-L1 pathway, and the combination of anti-VISTA and anti-PD-L1 synergize to promote anti-tumor immunity in the CT26 mouse colon tumor model[5]. The synthetic peptidomimetic CA170 is composed of L-serine, D-asparagine and L-threonine connected via diacylhydrazine and urea linker moieties[6]. It was designed to disrupt VISTA checkpoint signaling. A recent study using in vitro binding assays showed that there is no direct binding between CA170 and PD-L1, indicating that CA170 acts mainly via blocking VISTA immune checkpoint pathway[6]. Based on preclinical and clinical data (https://www.curis.com/pipeline/ca-170/), CA170 has excellent oral bioavailability, a relatively short half-life, and it can dose-dependently enhance the proliferation of T lymphocytes due to its ability to inhibit VISTA. In syngeneic murine tumor models, the antitumor effects of CA170 were similar to VISTA-blocking antibody. CA170 was well tolerated in phase I clinical trials and is undergoing phase II clinical trials for lung cancer, head and neck/oral cavity cancer, MSI-H positive cancers, and Hodgkin lymphoma (CTRI/2017/12/011026, ctri.nic.in)[7].

Immune checkpoints are antigen-independent secondary signals that modify the first signal provided by the interaction between peptide-MHC complexes and T-cell receptors (TCR), which confers specificity to the response[4]. Therefore, a combination of antigen-specific immunotherapy, such as peptide vaccination and immune checkpoint blockade should enhance both primary and secondary signaling events to increase antigen-specific T cell activation. We recently developed a multi-peptide KRAS vaccine (KVax) that contains several long peptides with high affinity and binds to multiple MHC class II alleles;[8] KVax is thus designed to promote KRAS-specific CD4+ T cell activation and infiltration to the tumor microenvironment which may facilitate tumor rejection[8]. KVax targets both the wild-type and G12D mutant forms of KRAS that are conserved between humans and mice. When KVax was administered in the inducible CCSP-KRAS murine lung cancer model before induction of mutant KRAS, a striking anti-tumor efficacy was observed[8].

The current study is the first to explore the efficacy of CA170, a small molecule antagonist of VISTA, for its anti-tumor effects in a carcinogen-induced murine primary lung tumor model. The effects of CA170 on immunity in the tumor microenvironment were also investigated using single-cell RNA-sequencing (scRNA-seq).

## Results

### CA170 treatment inhibits lung tumorigenesis in a primary mouse lung tumor model.
We tested whether a combination of KRAS vaccine plus CA170 could synergize to prevent lung tumorigenesis in the VC-induced primary lung tumor model. Six-week-old A/J mice were given a single i.p. injection of VC, and CA170 treatment was started one week after VC induction. The Kras peptide vaccine with STING agonist and AddaVax adjuvant was started two weeks after VC injection, followed by periodic boost vaccinations at the intervals specified in the Methods (Fig. 1A). Mice treated with either CA170 or KVax alone significantly decreased tumor multiplicity and tumor load (Fig. 1B). CA170 treatment decreased tumor load by 63% relative to the adjuvant control group (3.9 mm$^3$ vs. 10.6 mm$^3$, respectively); tumor load was also markedly lower in the KVax group (4.6 mm$^3$) relative to adjuvant control (Fig. 1B, right panel). Tumor load was further decreased by the KVax plus CA170 combination (2.2 mm$^3$), which represents ~80% inhibition relative to the adjuvant control and was significantly smaller than either single treatment alone groups.

### CA170 increased effector memory function of T cells and reduced G-MDSCs and Tregs in lung tumors.
We determined if the strong inhibition of tumor growth benefit observed in CA170/KVax-treated mice reflected activation of tumor-specific immune responses. CA170 significantly increased tumor-infiltrating CD8+ T cells (Fig. 2A, upper and lower left panel) with the phenotype that resembles effector-memory CD8+ T cells (Fig. 2B, upper panels). KVax and KVax plus CA170 (Combo) treatments increased tumor-infiltrating CD4+ T cells compared to adjuvant

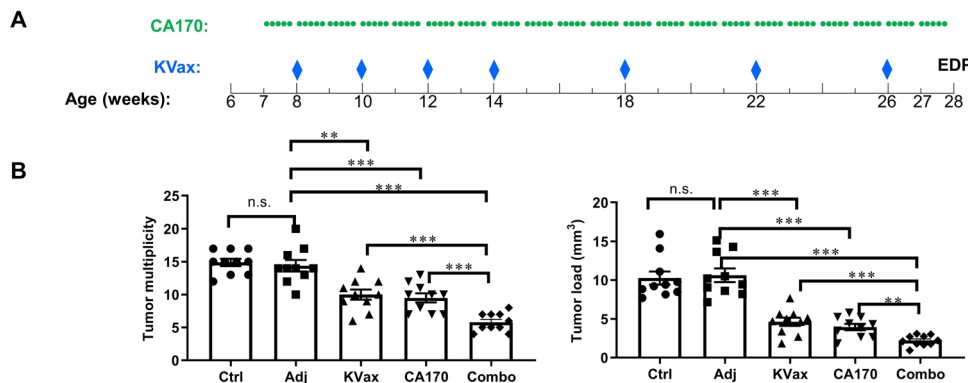

**Fig. 1 The combination of CA170 and KVax shows enhanced anti-tumor efficacy in VC-induced lung tumorigenesis. A** Experimental design outlining the timing of KVax vaccine treatment, CA170 administration, and experimental endpoint (EDP). **B** Tumor multiplicity and tumor load quantitation at the experimental endpoint. Data are shown as the mean ± SE, $n = 10$, *$P \leq 0.05$, **$P \leq 0.01$, ***$P \leq 0.001$ vs Ctrl.

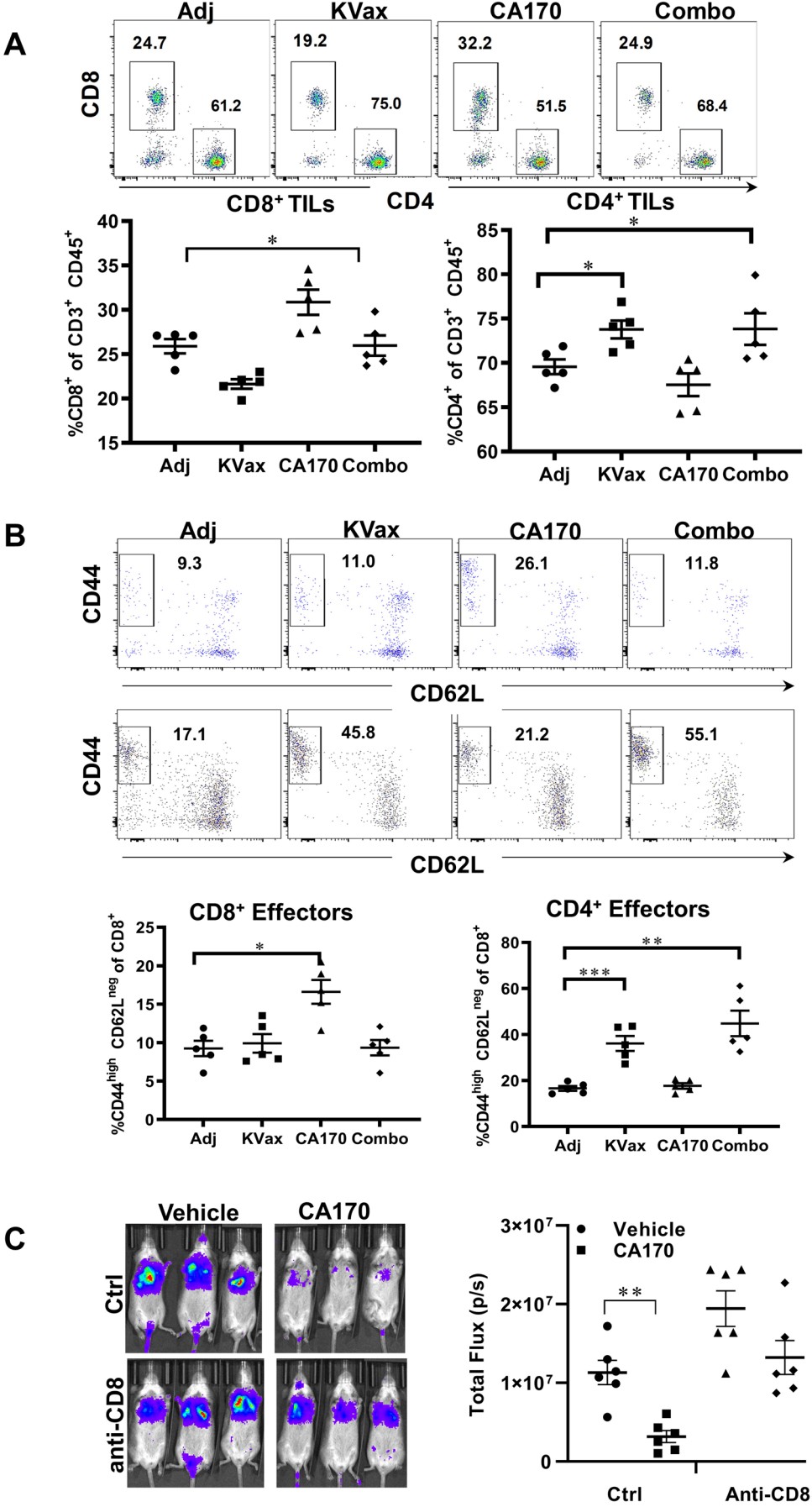

**Fig. 2 CA170 treatment promotes increased frequencies of tumor-infiltrating and effector memory CD8+ T cells.** Lung tumors were processed into single cells at the endpoint of the experiment and flow cytometric analysis was conducted on the tumor-infiltrating leukocytes isolated from each treatment group. **A** Representative (upper panel) and overall percentage of CD8+ (lower left panel) and CD4+ T cells (lower right panel) in TME. **B** Representative flow cytometry histograms (left panels) and combined results for staining of cells with CD44 and CD62L to assess effector cells from CD8+ (upper panels) or CD4+ T cells (lower panels). Data are shown as the mean ± SE, $n = 5$, *$P \le 0.05$, **$P \le 0.01$, ***$P \le 0.001$ vs Ctrl. **C** CA170 treatment depends on CD8+ T cells for the inhibition of lung tumor metastasis. Left panels: representative bioluminescence live imaging of lung tumor growth in Ctrl or CD8-depleting mice treated with vehicle or CA170. Right panel: quantitative data for bioluminescence imaging of the lung tumor growth of LKR13-Luc cells. Data are shown as the mean ± SE, $n = 6$, *$P \le 0.05$, **$P \le 0.01$ vs Vehicle.

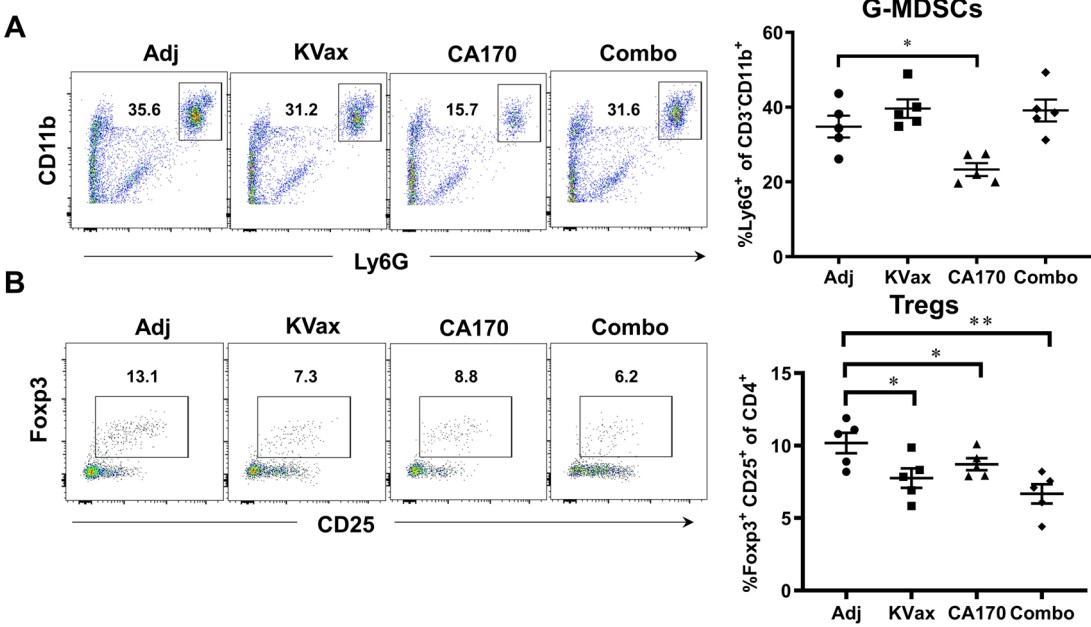

**Fig. 3 CA170 decreases tumor-infiltrating Tregs and G-MDSCs.** Lung tumors were processed into single cells, and the percent of G-MDSCs (**A**) and foxp3+CD25+ Tregs (**B**) were determined by flow cytometry. Data are shown as the mean ± SE, $n = 5$, *$P \le 0.05$, **$P \le 0.01$, ***$P \le 0.001$ vs Ctrl.

controls (Fig. 2A, upper and lower right panels), whereas no difference was observed in the CA170 alone treatment group. In the meantime, frequencies of CD44hi CD62Llow effector-memory CD4+T cells were enhanced in the lung tumors of KVax and Combo-treated mice (Fig. 2B, lower panels). To further evaluate the contributions of CD8+ T cells in CA170 mediated tumor inhibition, we tested CA170 in LKR13 lung metastasis syngeneic model upon CD8 depletion. In control animals, where CD8+ T cells are present, CA170 treatment significantly inhibited the metastasis of lung cancer, this inhibitory effect was decreased by CD8 depletion (Fig. 2C), as evidenced by the increase in the tumor burden in lung (Fig. 2C). These results suggest that CA170 promotes the recruitment and expansion of CD8 T cells inside tumors, whereas the KVax vaccine is largely responsible for the expansion of CD4 T cells within tumors.

Further, treatment with CA170 resulted in a significant reduction in percentages of G-MDSCs (Fig. 3). A reduction of Foxp3+ regulatory T cells (Tregs) within tumors was also seen in both KRAS vaccinated mice and CA170-treated mice; those treated with the combination of KRAS vaccine and CA170 had the most significant reduction of Tregs (Fig. 3). Consistent with increased tumor-infiltrating lymphocytes (TILs) in tumors, CA170 plus KVax treatment also increased the expression of IFN-γ, TNFα, and granzyme B in tumor-infiltrating CD4+ (Figs. 4A and C) and CD8+ (Figs. 4B and D) T cells in tumors. These data suggest that CA170 administration results in enhanced systemic anti-tumor immune responses in conjunction with the KRAS vaccine.

**Analysis of immune cell landscape in tumor tissues of experimental animals.** To uncover differential compositions of infiltrating immune cells in lung tumors from all five treatment groups, scRNA-seq was performed. A total of 80,000 CD45+ TILs in replicate samples from each treatment were analyzed. When the cells were grouped based on 20 principle components, they were categorized into 22 different clusters (Supplementary Fig. 1A, B), and the clusters were defined as different types of immune cells (Fig. 5A and B). Based on the expression of CD4 and CD8 genes, clusters 1 and 2 were defined as CD4+ T cell subsets, and cluster 11 was defined as a CD8+ T cell subset (Fig. 5C). Cell cycle gene expression was scored across all clusters, with cluster 15 exhibiting higher expression (Fig. S1C). Clusters 0 and 14 showed higher B cell-specific gene expression (Fig. S1D) and were thus denoted as B cell subsets. Based on the expression levels of myeloid-specific genes including *Itgam, Itgax, Fcgr3,* and *Ly6c2* (Fig. 5C, S1E), clusters 5, 7, and 13 were defined as dendritic cells; clusters 8, 9, and 12 as macrophages; and cluster 3 as neutrophils. Clusters 6 and 10 demonstrated NK cell and Treg cell gene expression, respectively (Fig. 5C). The percentage distribution of these different types of immune cells was calculated across the five different treatment groups. No significant changes were seen in percentages of different immune cell types for the adjuvant and CA170 treatments (Fig. 5D). However, as compared to control, both the KVax and combined CA170/KVax groups displayed an increased frequency of CD4 T cells and a decreased frequency of B cells (Fig. 5D).

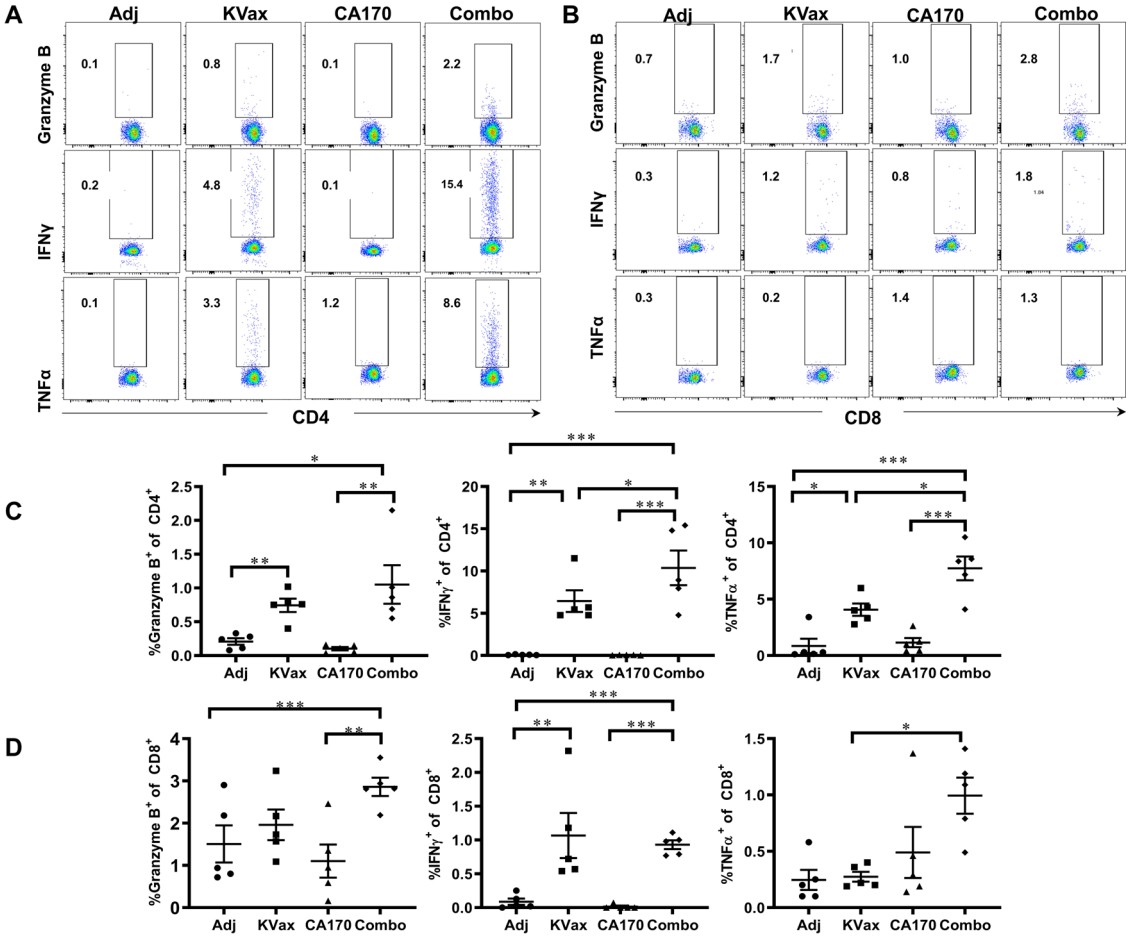

**Fig. 4 CA170 treatment enhances intracellular cytokine levels induced by Kvax in CD4⁺ and CD8⁺ T cells from lung tumors.** Lung tumors were processed into single cells at the endpoint of the experiment and flow cytometric analysis was conducted after stimulation with Kras peptide in the presence of monesin and brefeldin A. **A**, **B** Flow cytometric dot plots and **C**, **D** corresponding graphs depicting the amounts of cytokines granzyme B, IFNγ and TNFα measured by intracellular cytokine staining. Data are shown as the mean ± SE, $n = 5$, $*P \le 0.05$, $**P \le 0.01$, $***P \le 0.001$ vs Ctrl.

**KVax synergizes with CA170 treatment to increase antitumor effector CD4⁺ T cell responses and to decrease Treg-mediated immunosuppression in the tumor.** To better understand cellular heterogeneity of the CD4⁺ T cell subsets present in tumor tissue, scRNA-seq was performed on replicates of total lymphoid cells from tumors for each treatment group. For the analysis, these data were combined with the filtered CD4⁺ T cell data from the total immune cell sequencing in Fig. 5. Thus, a total of four CD4⁺ T cell datasets were analyzed together for each of the five treatment groups. Further analysis was performed on this combined dataset (~10,000 cells) using Canonical correlation analysis (CCA) and was based on 20 principle components from the 4 different CD4⁺ T cell clusters that were identified (Fig. 6A, Supplementary Fig. 2A). Cluster definition was based on published data for naïve, effector, effector-memory CD4 T cells and Treg cell gene expression profiles (Fig. 6C, Supplementary Fig. 2B–D). At the same time, Single R analysis was performed to confirm the cellular subset characterization based on published datasets of specific cell subsets. The percentage differential distribution of the CD4⁺ T cell subsets was calculated across the five treatment groups (Fig. 6D). A significant increase in the CD4⁺ effector cell subset (combined CD4⁺effector and CD4⁺effector-memory subsets) was observed for the KVax and combined CA170/KVax treatment groups, whereas a significant decrease in Treg subsets was found in these two groups as compared to the control group (Fig. 6E). Besides,

a significant increase in effector/Treg cell ratio was observed in the KVax and combined CA170/KVax treatment groups compared to the control. However, no such changes were noted in the adjuvant and CA170 treatment groups (Fig. 6F). Together, these observations suggest that KVax synergizes with CA170 treatment to increase antitumor effector CD4⁺ T cells response and to decrease Treg-mediated immunosuppression in the tumor.

**Effects of CA170 and KVax treatment on the function of CD8⁺ T cells in lung tumors.** Given that CA170 selectively blocks the VISTA negative checkpoint pathway, we investigated the effects of CA170, as well as the CA170 plus KVax combination treatment, on the phenotypic and functional heterogeneity of CD8⁺ T cells from lung tumor tissues. To do this, CD8⁺ T cell clusters from CD45⁺ scRNA-seq datasets and CD3⁺ scRNA-seq datasets were pooled into one Seurat object to simultaneously map the functional distinct CD8 subsets of control, adjuvant control, CA170, KVax, and CA170 plus KVax treatment groups (Fig. 7A). Unsupervised clustering analysis identified six CD8⁺ T cell populations when visualized by Uniform Manifold Approximation and Projection (UMAP) (Fig. 7B). The expression of signature genes (Supplementary Fig. 3A) and known functional markers (Fig. 7C) suggested clusters of naïve, effector, memory or exhausted CD8⁺ T cell subsets, as well as a few clusters that were

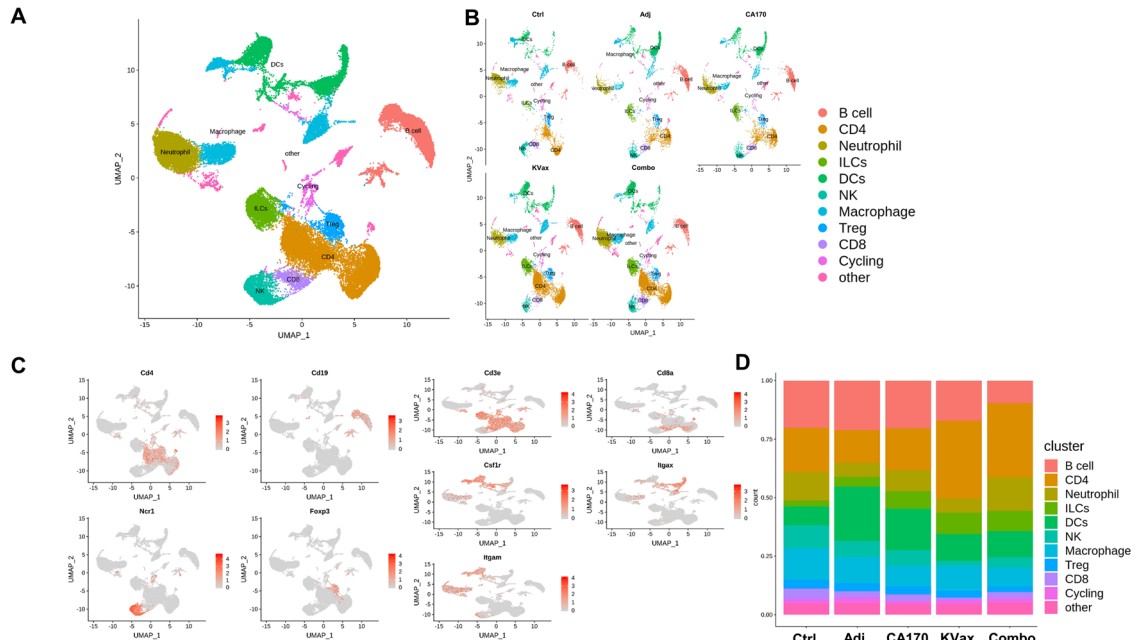

**Fig. 5 Immune cell heterogeneity for the various lung cancer treatments in mice.** Heterogeneity of total CD45[+] TILs from lung tumors all together (**A**) and separately under five different treatment conditions (Control, Adjuvant, CA170, KVAX vaccine and the combination of CA170 peptide plus KVAX vaccine) (**B**) in lung cancer shown by UMAP. **C** Feature plot shows representative genes for the different types of immune cells: CD4[+], CD8[+], B cells (Cd19), NK cells (Ncr1), Tregs (Foxp3), and Myeloid cell-specific genes (Cd3e, Itgam, Itgax, Csf1r) across all clusters for the five treatment conditions. **D** The bar plot shows the percentage of different immune cells under five different treatment conditions individually.

not well defined. Based on the expression of CD44, which is widely used as a T cell activation marker, six clusters could be separated into two major groups, naïve-like CD8 T cells (clusters 0 and 1) and antigen-experienced CD8[+] T cells (clusters 2, 3, 4, 5) (Supplementary Fig. 3B). Interestingly, clusters 0 and 1 expressed high levels of naïve CD8 T cell markers (*Lef1, Ccr7, Sell,* and *Dapl1*) as well as the tissue-resident marker *Itgae* (encode CD103) (Fig. 7C and Supplementary Fig. 3C). Among the antigen-experienced CD8[+] T cells, clusters 2 and 5 both expressed a high level of the T_RM marker gene *Cd69* (Fig. 7C). Notably, the frequency of T_RM (mainly cluster 2) was markedly increased in treatment groups when compared to the control group (Fig. 7D). Real-time RT-PCR with sorted CD8+ TILs also validated that CD69 expression is increased in CA170 treated lung tumors compared to untreated tumors (Fig. 7G). Cluster 3 was distinguished by high expression of chemokine receptor Cx3cr1 and molecules associated with cytotoxicity, such as Klrg1, Gzmb, Ifng, and Tbx21 (Fig. 7C and Supplementary Fig. 3D). In contrast, cluster 4 showed low expression of effector molecules when compared to cluster 3, while expressing high levels of multiple inhibitory receptors, including Pdcd1, Lag3, Havcr2, and Ctla4 (Fig. 7C and Supplementary Fig. 3D), and exhaustion related transcription factors including Eomes, Irf4, Tox and Nr4a family (Fig. 7C). The transcriptional levels of Prf1, Gzma, and Cx3cr1 were all upregulated in CD8[+] TILs treated with CA170 (Fig. 7G upper left panel), whereas mRNA levels of these genes remain unchanged in the CD4+ TILs (Fig. 7G upper right panel) isolated from the same CA170 treated lung tumors. Lag3 and Pdcd1 were both slightly decreased in CD8+ TILs, but not significantly (Fig. 7G lower left panel). We next sought to test the effects of CA170 and the CA170/KVax combination treatments on the ratio of effector CD8[+] T cells to exhausted CD8[+] T cells (effector/exhaustion ratio). CA170, KVax and the combined treatment each resulted in increased effector/exhaustion ratios compared to the control and adjuvant groups (Fig. 7E). Furthermore, the expression of inhibitory receptors in exhausted

CD8[+] T cells was decreased with CA170 and the combined CA170/KVax treatment compared to the adjuvant and KVax treatment groups (Fig. 7F). In summary, these results suggest that CA170 synergizes with KVax to reduce the signals that can suppress CD8[+] T cells.

## Discussion
Augmenting the immune system to eradicate cancer dates back at least a century. A major advance in cancer immunotherapy has occurred over the past decade with the use of monoclonal antibodies to block immune checkpoints to enhance antitumor immunity. Launch of the first CTLA-4 blocking antibody (ipilimumab) in 2010[9] and PD-1/PD-L1 blocking antibodies (pembrolizumab and nivolumab) in 2014[10,11] represented watershed moments that showed the promise of using immune checkpoint inhibitor (ICIs) as a standard of immunotherapy for many tumor types[12]. Currently, more than ten immunomodulating antibodies have received regulatory approval worldwide and thousands of others are in active clinical trials[13]. However, antibody-based therapies suffer from several shortcomings that limit their application. The large size of antibodies prevents them from penetrating deep into tumors, thus reducing their ability to mediate antibody dependent cell-mediated cytotoxicity (ADCC)[14]. Additionally, Fc regions of antibodies induce complement-dependent cytotoxicity (CDC) effects, which can impair immune cells and could be one of the reasons to induce immune-related adverse events (irAEs)[15]. The relatively long half-life of antibodies increases the difficulty in drug clearance when side effects occur[15]. Therefore, there is an urgent need to develop non-antibody drugs that effectively target immune checkpoints.

Compared to monoclonal antibodies, immunotherapy with small molecules has several advantages including low cost, ease of manufacturing, optimal pharmacokinetics, the potential for oral administration, flexible clinical dosing, relatively short half-life, and the potential for lower systemic toxicity. More importantly, small molecule ICIs not only can target immune suppressive mechanisms, similar to antibodies, but they can also stimulate

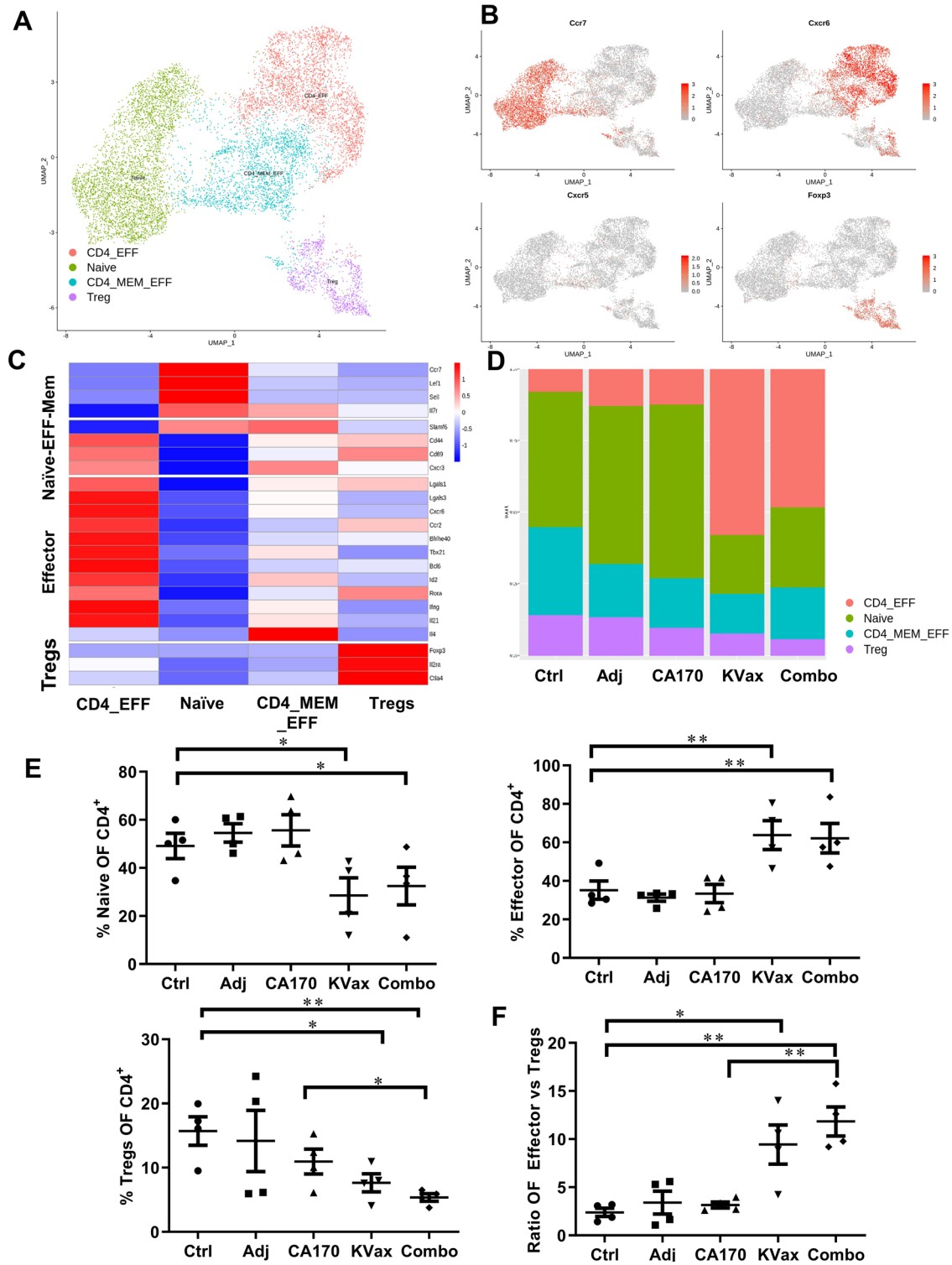

**Fig. 6 KVax synergizes with CA170 treatment to increase antitumor effector CD4+ T cells response and to decrease Treg-mediated immunosuppression in the tumor. A** UMAP plot showing cellular heterogeneity of CD4+ T cells from two CD45+ scRNA-seq datasets and CD3+scRNA-seq datasets. Each dot represents a single cell and is colored according to corresponding CD4 subset. **B** Representative genes for naïve (Ccr7), Effector Th1 and T_FH (Cxcr6 and Cxcr5, respectively) and Treg (Foxp3) specific genes shown across all clusters using the feature plot. **C** Heatmap showing average expression of different CD4+ Naïve and Effector memory T cell-specific genes (Naïve-Eff-Mem), CD4 Effector T cell-specific genes and Treg-specific genes across all cells in all clusters for the five lung cancer treatments. **D** Bar plot showing the percentage of different CD4 T cell clusters for the five treatment conditions individually. **E** Percentage of naïve, effector, and suppressor (Treg) cells across four CD4+ T cell samples for the five different lung cancer treatments. **F** Ratio of CD4 Effector vs Suppressor cells across four CD4+ T cell samples for the five different lung cancer treatments. Data are shown as the mean ± SE, $n = 4$, *$P \leq 0.05$, **$P \leq 0.01$, ***$P \leq 0.001$ vs Ctrl.

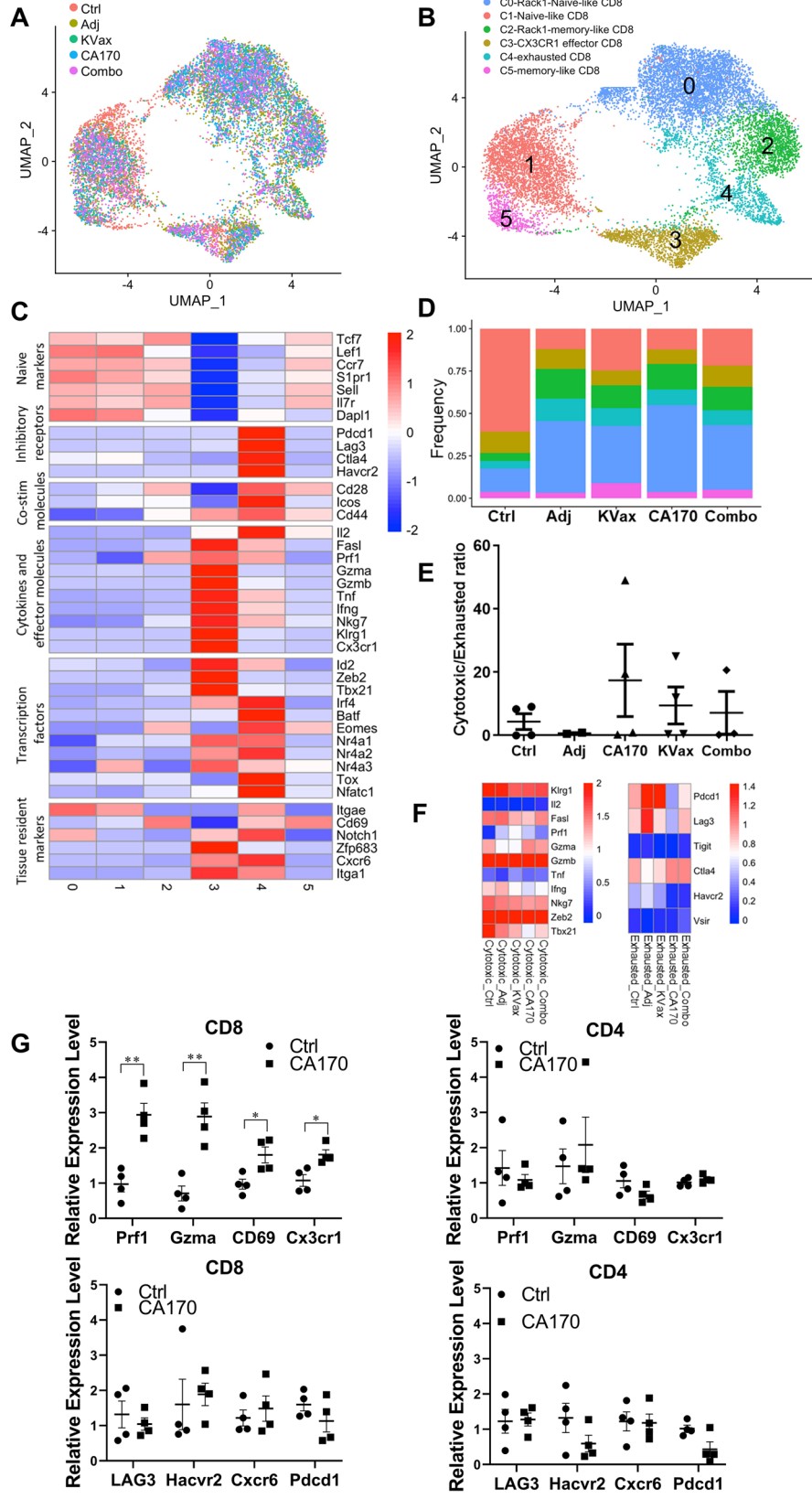

**Fig. 7 CA170 treatment increases cytotoxic/exhausted CD8$^+$ ratio. A** The UMAP projection of CD8 T cells from two CD45$^+$scRNA-seq datasets and CD3$^+$scRNA-seq datasets. Each dot corresponds to one single cell and is colored according to its treatment. **B** The UMAP projection showing the formation of 6 main clusters, including 2 naïve-like CD8$^+$ T cells, 2 T$_{RM}$-like CD8$^+$ T cells, 1 effector population and 1 exhausted population. Each dot corresponds to one single cell and is colored according to its cell type. **C** Z-score normalized mean expression of selected T cell function-associated genes in each cell cluster. **D** Stacked percentage bar plot showing the proportion of each cell type in the 5 treatment groups. Colors represent various cell types as shown in (**B**). **E** Dotplot showing the statistical comparison of the frequencies of cytotoxic/exhausted CD8$^+$ T cells in the different treatment groups. Data are shown as the mean ± SE, n = 4, *$P \leq 0.05$, **$P \leq 0.01$, ***$P \leq 0.001$ vs Ctrl. **F** Z-score normalized mean expression of cytotoxicity-related molecules and inhibitory receptors in Cx3cr1$^+$ cytotoxic and exhausted CD8$^+$ T cells, respectively. Each column represents a different treatment. **G** Real-time RT-PCR validation of genes regulated by CA170 in CD4$^+$ or CD8$^+$ TILs isolated from tumors treated with vehicle or CA170. Data are shown as the mean ± SE of three replicate samples per group, n = 4, *$P \leq 0.05$, **$P \leq 0.01$ vs Ctrl, one-way ANOVA.

intracellular pathways downstream of immune checkpoints in innate or adaptive immune cells that antibodies are unable to access[12]. Small molecule ICIs thus provide an alternative treatment modality that could be used alone or in concert with immune checkpoint-blocking B antibodies to address the issues of suboptimal clinical response and/or drug resistance[16,17]. Our study represents the first test of CA170 in a carcinogen-induced murine primary lung tumor model, which bears similar morphology, histopathology, and molecular anomalies as those observed in human tumors. In this model, orally administered CA170 caused no apparent toxicities including body weight losses and other visible signs of toxicities, and was able, by itself, to reduce tumor burden by >60%. Notably, 200 mM CA170 did not directly kill cancer cells (Supplementary Fig. 4), which is far above the in vivo levels achieved with a standard physiologic oral dose (10 mg CA170 per kg body weight). When CA170 was combined with a peptide vaccine that specifically targets *Kras*, which is the driver gene in VC-induced tumors, tumor inhibition was almost 80% when compared to adjuvant-treated controls (Fig. 1). Our results are consistent with markedly greater potency and efficacy using the combined CA170/KRAS vaccine treatment in inhibiting tumor growth versus the KRAS vaccine alone. This study constitutes a critical step toward the clinical translation of CA170 for preventing lung cancer.

A recent study using in vitro binding assays showed that there is no direct binding between CA170 and PD-L1[6], suggesting that CA170 may act mainly via blocking VISTA immune checkpoint pathway rather than on PD-L1/PD-1 pathway. In the current study, we systematically analyzed immune responses and immune cell subset changes in response to CA170 and KRAS vaccine treatment using both flow cytometry and scRNA-seq. Interestingly, CA170 significantly increased tumor-infiltrating CD8$^+$ T cells (Fig. 2A), and enhanced effector-memory T cell frequencies and function for both CD4$^+$ and CD8$^+$T cells. These changes coincided with significant reductions in G-MDSCs and Foxp3$^+$ Treg populations within tumors (Fig. 3). scRNA-seq also confirmed that the myeloid cell clusters were remodeled by CA170 and by the combination of CA170 plus KRAS vaccine. CD8$^+$ tumor-infiltrating lymphocytes play a critical role in antitumor immune responses through recognition of tumor antigens and direct killing of tumor cells[18]. However, CD8$^+$ T cells in growing tumors are often functionally impaired as a result of suppressive signals from the tumor microenvironment, which can result in T cell exhaustion and a failure to effectively eliminate cancer cells[19]. A greater density of tissue-resident memory T (T$_{RM}$) cells has been linked to better lung cancer survival outcomes[20]. Indeed, real-time RT-PCR analysis demonstrated that T$_{RM}$ marker gene *Cd69* was upregulated in CD8$^+$ TILs isolated from CA170 treated lung tumors compared to untreated tumors (Fig. 7G). CX$_3$CR1$^+$ CD8 T cells were recently identified as an effector subset with potent cytolytic function against chronic viral infection and cancer[21,22]. The transcriptional levels of Cx3cr1 and other key genes for cytolytic function, such Prf1 and Gzma, were all upregulated in CD8$^+$ TILs treated with CA170

(Fig. 7G upper left panel). Of note, a significant increase in effector/Treg ratio was observed in the KVax and combined CA170/KVax treatment groups compared to control (Fig. 7E). The KVax vaccine also increased tumor-infiltrating CD4$^+$ T cells (Fig. 2A), decreased Tregs, and significantly increased a CD4$^+$ effector cell subset (Fig. 3) as well as the ratio of this subset to Tregs (Figs. 3 and 6E). These results suggest that CA170 can work with KVax to blunt the suppressive signals on CD8$^+$ T cells.

T cell costimulatory and coinhibitory pathways have much broader immunoregulatory functions by controlling effector, memory and Treg cells, as well as naïve T cells. A recent study classified naïve CD4$^+$ T cells into six clusters and found that within the naïve T population, deficiency of VISTA primed cells toward an activated/memory state[23]. In line with the known functions of T cell costimulatory/coinhibitory pathways, our studies indicate that CA170 administration enhances systemic anti-tumor immune responses. The data suggest that CA170 drives an adaptive T cell tumor-specific immune response resulting in lung tumor regression. Importantly, CA170 does not appear to promote the development of immune-associated toxicities. In preclinical safety studies conducted in rodents and non-human primates, orally administered CA170 showed no signs of toxicity when dosed up to 1000 mg/kg for 28 consecutive days[24,25]. CA170 has high oral bioavailability in both mice and monkeys, with a plasma half-life of ~3.5 h (mouse) and 3.25–4.0 h (monkey)[26]. In a recent clinical trial in patients with advanced solid tumors or lymphomas, CA170 had a favorable safety profile (up to 1200 mg BID) and a short half-life (comparing to antibody ICIs) and showed immune-modulating effects with tumor regression[26]. A recent study further confirmed that CA170 selectively inhibits PD-L1/L2 and VISTA pathways, and exhibits significant anti-tumor efficacy in murine colon and melanoma tumor models[27].

In summary, no small molecule immunotherapy has yet been approved for cancer immunotherapy, CA170 is the only small molecule ICIs that has been announced to move to Phase III clinical trials for multiple cancers including lung cancer. Our data support advancing the use of CA170 for lung cancer, and it indicates the potential of combining CA170 with the KRAS vaccine to achieve even better anti-tumor efficacy in lung cancer patients. Finally, this combined treatment modality could represent an exciting approach for the immunoprevention of lung cancer in those at increased risk.

## Methods

**Vaccine preparation and immunization.** The Kras peptide vaccine KVax contains four peptides that represent different regions of the Kras protein: Kras-G12D (KLVVVGADGVGKSALTI), Kras-61Wt (KLVVVGAGGVGKSALTI), Kras-63Wt (SALTIQLIQNHFVDE) and Kras-68Wt (FLCVFAINNTKSFED). These peptides were synthesized by Genemed Synthesis Inc. (San Antonio, TX) and their purity (>95%) was verified by HPLC. At each vaccination time, mice were given KVax s. c. in the flank containing 50 µg of each peptide. Adjuvants used in the study included the STING agonist c-di-AMP (each delivered at 10 µg/mouse) and AddaVax (50 µL/mouse). The total vaccine volume was 100 µL/mouse.

**Mouse models and treatments**. Female A/J mice were purchased from the Jackson Laboratory. Mice were kept in the Biomedical Resource Center at the Medical College of Wisconsin, Milwaukee, WI, and all procedures were approved by the Institutional Animal Care and Use Committee (IACUC).

Lung cancer is a heterogeneous disease comprised of multiple histologic subtypes that harbor distinct biology and immune profiles[28]. Our mouse model of primary lung carcinogenesis represents a valuable tool for the study of tumor initiation, promotion, and therapy. Carcinogen-induced lung tumor models bear similar morphology, histopathology, and molecular anomalies to those observed in human tumors. Vinyl carbamate (VC) primarily induces Kras mutation and leads to the formation of lung adenomas and adenocarcinomas in humans[29]. In these tumors, an immunosuppressive microenvironment[30] may hamper anti-cancer treatments including vaccines.

Five- to six-week-old female A/J mice were divided into five groups: (a) control mice that received PBS; (b) adjuvant control mice that received a STING agonist plusAddaVax; (c) KVax mice treated with KVax plus adjuvant; (d) CA170 treatment (10 mg/kg in PBS, Curis, Inc.) by oral gavage five times per week; and (e) combined treatment with KVax/adjuvant plus CA170. The experimental treatment scheme is shown in Fig. 1A. Mice were administered one dose of the carcinogen vinyl carbamate[31] (VC; Toronto Research Chemicals, Inc.) by i.p. injection (16 mg/kg in sterile saline). CA170 treatment was started one week after VC induction of lung adenocarcinoma. KVax was started two weeks after VC induction, followed by four more vaccinations at 2-week intervals, and additional boost vaccines every four weeks for the duration of the experiments.

Immediately after euthanasia, lungs from the various treatment groups were harvested, inflated, formalin-fixed overnight and then stored in 70% ethanol. Tumor multiplicity was determined by surface tumor counting as previously described using a dissecting microscope. Tumor volume for each tumor was calculated and tumor load was calculated as the sum of tumor volume from each animal using a dissecting microscope, as previously described[32].

For CD8 depletion, LKR13 cells expressing luciferase (LKR13-Luc) were inoculated by tail vein injection to 8 weeks old sv129 mice, CD8 mAbs (clone-2.43, 250 µg, BioXcell, BP0061) was i.p. injected on 1 day before and 1 day after tumor inoculation and repeated once per week. Eight days after the first i.p. injection, PBMCs were collected and assayed for CD4+ and CD8+ T cell population using flow cytometry. Tumor growth was monitored by IVIS Spectrum.

**In vitro proliferation assay**. LKR13 cells originated from Kras^LA1 mice, and express mutant Kras^G12D on the sv129 background, were a generous gift from Dr. Jonathan M. Kurie (MD Anderson), and were cultured in RPMI-1640 (Thermofisher, 11875-093) supplemented with 10% FBS and 1% penicillin/streptomycin. For proliferation assay, 2,000 cells per well were seeded in 96-well flat-bottom plate, 24 h later, cells were exposed to various concentrations of CA170 ranging from 0.78 to 200 µM, whereas control groups wells received fresh complete medium. Plates were incubated at 37 °C under 5% $CO_2$ and monitored for real-time cell confluence data every two hours using the IncuCyte Live Cell analysis system (Essen Bioscience, Ann Arbor, MI), and proliferation was analyzed based on real-time confluency data with the IncuCyte 2011A software. All assays were performed in triplicate or quadruplicate.

**Flow cytometry**. For immune profiling, tumors were harvested and pooled from each mouse at the end of the study, minced into 1–2 mm pieces and digested at 37 °C for 20 min with mouse tumor dissociation buffer (MiltenyiBiotec, CA) to generate single cell suspensions per the manufacturer's instructions. Tumor-infiltrating leukocytes were directly stained for flow cytometry sorting or analysis.

Isolated cells were stained with cell surface and viability markers first: 7AAD for live/dead cells, BV786 anti-CD45, APC eFluro780 anti-CD3, FITC anti-CD4, BUV396 anti-CD8a, SB600 anti-CD19, PE-Cy7 anti-CD44, APC anti-CD62L, and PE anti-CD25 antibodies. For intracellular cytokine staining, cells were stimulated for 4 h in RPMI medium containing Kras peptides, 10% FBS, 2 mM L-glutamine, 50 µM 2-mercaptoethanol, 1% penicillin–streptomycin, 1× monensin, 1× brefeldin A (Thermofisher Sci). For foxp3 staining, cells were then washed, fixed and permeabilized and stained with Foxp3/Transcription Factor eFluor450 Staining Buffer sets (Thermofisher) following the manufacturer's instructions. For intracellular cytokine analysis, cells were fixed with 2% paraformaldehyde, permeabilized with 0.5% saponin, and stained with intracellular cytokine staining buffer containing brefeldin A, APC anti-granzyme B, PE anti-IFN-γ and PE-Cy7 anti-TNF-α antibody, and then analyzed by flow cytometry. T cells stained with isotype control antibody were used as negative controls. To detect MDSC in tumors, cells were stained with PerCP-Cy5.5 anti-CD45, FITC anti-CD11b, APC anti-CD11c, PE anti-Ly6G and PE-Cy7 anti-Ly6C Ab. Flow cytometry was conducted using an LSR Fortessa™ X-20 or LSR-II flow cytometer (Becton Dickinson). Data were analyzed using FlowJo software.

**Single-cell RNA sequencing**. For scRNA-seq, VC-induced primary lung tumors were harvested and pooled from different treatment groups at the end of the study, then minced and digested at 37ºC for 20 min with mouse tumor dissociation buffer (MiltenyiBiotec, CA) to generate single cell suspensions per the manufacturer's instructions. Tumor-infiltrating leukocytes were directly stained with 7-AAD, CD45, and CD3 surface markers, and CD3+ or CD45+ TILs were flow-sorted.

Flow sorted TILs were spin down at $300 \times g$ for 5 min, counted manually with Neubauer Chamber. About $1.6 \times 10^4$ cells were loaded onto the 10X Chromium Controller per the manufacturer's instructions, resulting in recovery of about $1 \times 10^4$ cells. The scRNA-seq libraries were generated by Chromium Single Cell 3′ v3 Reagent Kits (10× Genomics) and sequenced using NextSeq 500/550 High Output Kits v2 (150 cycles) (Illumina) according to the manufacturer's protocol. There were two replicates for each experimental group.

**Single-cell RNA sequencing data analysis**. Raw sequencing data were de-multiplexed and converted to gene-barcode matrices using the Cell Ranger (version 2.2.0) mkfastq and count functions, respectively (10× Genomics). The mouse reference genome mm10 was used for alignment. Data were further analyzed in R (version 3.4.0) using Seurat (version 3). The number of genes detected per cell, number of unique molecular identifiers (UMIs), the percent mitochondrial genes were plotted, and outliers were removed (cells that expressed less than 200 and more than 2500 genes, and cells with >0.05 percent mitochondrial genes) to filter out doublets and dead cells. Differences in number of UMIs and percent mito-chondrial reads were regressed out. Raw UMI counts were normalized and log-transformed. For analyzing CD45+ cells, 10 single-cell RNA sequencing datasets from 5 experimental groups were loaded into one Seurat object. Integrated analysis was then performed to identify shared cell states that are present across different datasets. The same integrated analysis was performed for CD3 datasets. Principal component analysis was performed using variable genes, and the top 20 most statistically significant principal components were used for t-SNE analysis. To identify marker genes, the FindAllMarkers function was used with the likelihood-ratio test for single-cell gene expression. For each cluster, only genes that were expressed in more than 25% of cells with at least 0.25-fold difference were considered.

**Quantitative reverse transcription PCR**. Total RNA was extracted from CD4+ or CD8+ TILs isolated from tumors treated with vehicle (Control) or CA170 using RNA Plus Micro kits (Qiagen). cDNA was synthesized with iScript kits (Qiagen), and quantitative reverse transcription PCR (qRT–PCR) conducted using SsoAd-vanced Universal SYBR Green (Qiagen) per the manufacturer's instructions. Primers used in qRT-PCR were as follows:

Prf1 (forward, CTGCCACTCGGTCAGAATG; reverse, CGGAGGGTAGTCA CATCCCAT);

Gzma (forward, GGGGCTCACTCAATCAATAAGG; reverse, CATCCTGCT ACTCGGCATCT);

CD69 (forward, CCCTTGGGCTGTGTTAATAGTG; reverse, AACTTCTCGT ACAAGCCTGGG);

Cx3cr1 (forward, GAGTATGACGATTCTGCTGAGG; reverse, CAGACCGAA CGTGAAGACGAG);

Pdcd1 (forward, CAGCTTGTCCAACTGGTCG; reverse, GCTCAAACCATTA CAGAAGGCG);

Lag3 (forward, CTGGGACTGCTTTGGGAAG; reverse, GGTTGATGTTGCC AGATAACCC); Havcr2 (forward, TCAGGTCTTACCCTCAACTGTG; reverse, GGCATTCTTACCAACCTCAAACA);

Cxcr6 (forward, GAGTCAGCTCTGTACGATGGG; reverse, TCCTTGAACTT TAGGAAGCGTTT).

**Statistics and reproducibility**. All in vitro assays were performed at least in triplicate. Five to nine mice per group were used for the in vivo studies. General statistical analyses were performed using GraphPad Prism 7.0 software. Unpaired 2-tailed Student's t-tests or 1-way ANOVA were used for a column, multiple columns, and group analyses, respectively. *$P < 0.05$, **$P < 0.01$, and ***$P < 0.001$ were considered as statistically significant.

**Reporting summary**. Further information on research design is available in the Nature Research Reporting Summary linked to this article.

## Data availability

The raw and processed total RNA and sc-RNA sequencing data files have been deposited in NCBI's Gene Expression Omnibus and are accessible through GEO Series accession numberGSE176091. The source data underlying all figures are provided as Supplementary Data 1. All other relevant data are available in the article, supplementary information, or from the corresponding author upon reasonable request.

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

## Acknowledgements

We are grateful to Dr. Jonathan M. Kurie (MD Anderson) for providing the LKR13 cells, and to Curis, Inc. for providing CA170. This work was funded by the National Cancer Institute (HHSN26100002, 75N91019D00020-E01, and 75N91019D00020-E04). Drs. Shizuko Sei, Robert H. Shoemaker, and Ronald A. Lubet from NCI participated in the study design and the manuscript editing. This work was also supported by the grants from National Institutes of Health [R01CA223804 (M.Y.; L.W.); R01CA232433 (M.Y.); R01CA223804 (M.Y.); AI125741 (W.C.); R01CA164225 (L.W.); R01AI148403 (W.C.)], America Cancer Society Research Scholar Grant RSG-18-045-01 - LIB (L.W.), and America Cancer Society Research Scholar Grant (W.C.).

## Author contributions

M.Y. and J.P. were responsible for the overall experimental design. L.W. contributed to the conceptual design and provided key reagents for this study. Q.Z. and J.P. assessed anti-cancer efficacy in animal model. K.P. and J.P. conducted flow cytometry analysis, Y. C., J.P., A.K., and G.X. conducted single-cell sequencing studies. The following were largely responsible for writing, reviewing, and editing the manuscript: J.P., Q.Z., Y.C., A. K., C.M., B.J., L.W., S.S., R.H.S., R.A.L., Y.W., and M.Y.

## Competing interests

M.Y. is a co-founder of OncoC4, Inc. All other authors declare no competing interests.
