## [Peer Review File · Communications Biology]

Reviewers' Comments:

Reviewer #1:

Remarks to the Author:

The manuscript reports a very well conducted preclinical work analyzing a small molecule antagonist of VISTA, ca170, in lung cancer models. The work very well addresses the current interest in immune checkpoint inhibitors other than antibodies, as well as the potential clinical impact of inhibiting VISTA as immune checkpoint. The findings, especially regarding the influence on immune cells, are interesting and may have further research impact.

Here my suggestions:

1. There is confusion throughout the paper between Immune checkpoints (ICs) - the target- and immune checkpoint inhibitors (ICIs) - the drugs. Please revise the manuscript with caution on this aspect
2. CTLA-4, PD-1, Lag-3, Tim-3, TIGIT and VISTA should be defined
3. the sentence "CA170 exhibits antitumor effects the same as that obtained by VISTA-blocking antibody" is not well expressed and should be rephrased.
4. The last paragraph of the introduction section reports methods and results: in my opinion only the first sentence of the paragraph should be part of the introduction.
5. Results: first paragraph is a comment on methods. It should be described in the methods section
6. Subheading "CA170 increased effector memory function of T cells and reduced G-MDSCs and Tregs in lung tumors": these results are about immune infiltrate, therefore the conclusion of the paragraph should not infer an impact on tumor regression as a matter of fact
7. Subheading "Effects of CA170 and KVax treatment on the heterogeneity and function of CD8+ T cells in lung tumors": this section is including comments that should be part of the discussion and not of the results ("A greater density of tissue-resident memory T (TRM) cells has been linked to better lung cancer survival outcomes"; "Of note, CD8 T cells were recently identified as an effector subset with potent cytolytic function against chronic viral cancer and infection")
8. In the discussion, the authors state that complement dependent cytotoxicity (CDC) effects, can impair immune cells and induce immune-related adverse events: CDC is not the only mechanism of irAEs, and this affirmation could generate confusion. Please clarify.

Reviewer #2:

Remarks to the Author:

This work by Pan et al claimed the role of tripeptide small molecule antagonist of VISTA CA170 on immune cells by using lung cancer mouse models. Overall, this study lacks novelty (please see refs 1,2) and its molecular mechanism remains unclear.

Below are my concerns:

- 1) It will be interesting to validate some potential genes mediated by VISTA in their lung cancer models according to their single cell RNA-seq data. Molecular biology analysis could be done, such as Western blot and qPCR.
- 2) Rescue experiments might be done to further examine the contributions by various immune cell subsets including effector MDSCs, CD4+ T cells, CD8+ T cells and Treg cells in CA170 mediated lung cancer tumorigenesis.

References

1. Small molecule inhibitors targeting the PD-1/PD-L1 signaling pathway. Acta Pharmacol Sin. 2020 Mar 9. doi: 10.1038/s41401-020-0366-x.
2. Excellent CBR and prolonged PFS in non-squamous NSCLC with oral CA-170, an inhibitor of VISTA and PD-L1. DOI: <https://doi.org/10.1093/annonc/mdz253.035>.

Thus, my recommendation is Reject.

Reviewer #3:

Remarks to the Author:

The manuscript by Pan et al showed that "VISTA antagonism by CA170 revealed strong efficacy against lung tumorigenesis with broad immunoregulatory functions that influence effector, memory and regulatory T cells, and drives an adaptive T cell tumor-specific immune response that enhances efficacy of the Kras vaccine."

The paper was carry out with a good pre-clinical methodology which could lead to a substantial rational for phase I trial desing.

Reviewer #4:

Remarks to the Author:

the manuscript is satisfactory no changes is needed

Reviewers' comments:

Reviewer #1 (Remarks to the Author):

The manuscript reports a very well conducted preclinical work analyzing a small molecule antagonist of VISTA, ca170, in lung cancer models. The work very well addresses the current interest in immune checkpoint inhibitors other than antibodies, as well as the potential clinical impact of inhibiting VISTA as immune checkpoint. The findings, especially regarding the influence on immune cells, are interesting and may have further research impact.

Response: We thank this reviewer for these positive comments.

Here my suggestions:

1. There is confusion throughout the paper between Immune checkpoints (ICs) - the target- and immune checkpoint inhibitors (ICIs) - the drugs. Please revise the manuscript with caution on this aspect.

Response: We have clarified ICs versus ICIs in the revision.

2. CTLA-4, PD-1, Lag-3, Tim-3, TIGIT and VISTA should be defined.

Response: We have defined all these terms in the revision.

3. the sentence "CA170 exhibits antitumor effects the same as that obtained by VISTA-blocking antibody" is not well expressed and should be rephrased.

Response: We rephrased the sentence in the revision. The rephrased sentence states "the antitumor effects of CA170 were similar to VISTA-blocking antibody".

4. The last paragraph of the introduction section reports methods and results: in my opinion only the first sentence of the paragraph should be part of the introduction.

Response: We revised the last paragraph of the introduction section.

5. Results: first paragraph is a comment on methods. It should be described in the methods section

Response: We moved the first paragraph in the method section.

6. Subheading "CA170 increased effector memory function of T cells and reduced G-MDSCs and Tregs in lung tumors": these results are about immune infiltrate, therefore the conclusion of the paragraph should not infer an impact on tumor regression as a matter of fact

Response: We corrected the conclusion of the paragraph.

7. Subheading "Effects of CA170 and KVax treatment on the heterogeneity and function of CD8+ T cells in lung tumors": this section is including comments that should be part of the discussion and not of the results ("A greater density of tissue-resident memory T (TRM) cells has been linked to better lung cancer survival outcomes"; "Of note, CD8 T cells were recently identified as an effector subset with potent cytolytic function against chronic viral cancer and infection")

Response: We have moved those sentences into the discussion section.

8. In the discussion, the authors state that complement dependent cytotoxicity (CDC) effects, can impair immune cells and induce immune-related adverse events: CDC is not the only mechanism of irAEs, and this affirmation could generate confusion. Please clarify.

Response: It is our understanding that the short half-lives of Small-Molecule Drugs such as CA-170 is one of the reasons for lowering their immune-related adverse effects. In other words, better control of their pharmacodynamics and pharmacokinetics will help to prevent irAEs. In general, the irAEs were caused by activation of autoreactive T cells, in some cases autoantibodies, due to the reduced activation threshold of antigen receptor signaling. Tumor-targeting antibodies will require the Fc regions to initiate ADCC/CDC to kill tumor cells. However, the Fc regions of immune checkpoint blocking antibodies will be modified to prevent ADCC/CDC. Otherwise, CD8+ tumor-infiltrating lymphocytes will be eliminated via ADCC/CDC in an Fc-dependent manner. Therefore, this may not be the cause of irAEs by ICIs.

Reviewer #2 (Remarks to the Author):

This work by Pan et al claimed the role of tripeptide small molecule antagonist of VISTA CA170 on immune cells by using lung cancer mouse models. Overall, this study lacks novelty (please see refs 1,2) and its molecular mechanism remains unclear.

References

1. Small molecule inhibitors targeting the PD-1/PD-L1 signaling pathway. *Acta Pharmacol Sin.* 2020 Mar 9. doi: 10.1038/s41401-020-0366-x.

2. *Excellent CBR and prolonged PFS in non-squamous NSCLC with oral CA-170, an inhibitor of VISTA and PD-L1.*

DOI:<https://doi.org/10.1093/annonc/mdz253.035>.

Response: Thank you for providing these additional references, which we have added to the revised manuscript. There are multiple novel aspects of our studies. This is the first test of the efficacy of CA170 in a carcinogen-induced primary lung tumor model. This may benefit and inform future studies on its use for cancer prevention. We also investigated the effects of CA170 on immunity in the tumor microenvironment using scRNA-seq. We found that CA170 significantly decreased Tregs and G-MDSCs in the tumor microenvironment and resulted in enhanced T cell function. Based on the reviewer's suggestion, we also used qPCR to validate some potential genes identified by scRNA-seq (see comment 1 below).

Below are my concerns:

1) It will be interesting to validate some potential genes mediated by VISTA in their lung cancer models according to their single cell RNA-seq data. Molecular biology analysis could be done, such as Western blot and qPCR.

Response: We agree with this reviewer. We have now validated some potential genes by qPCR and have added these new data in the revised manuscript.

2) Rescue experiments might be done to further exam the contributions by various immune cell subsets including effector MDSCs, CD4+ T cells, CD8+ T cells and Treg cells in CA170 mediated lung cancer tumorigenesis.

Response: We agree with the reviewer that further examining the contributions of these subsets of immune cells is important to understand the mechanisms of CA170. Since depleting Tregs with anti-CD25 might also deplete other CD25 expressing populations, a clearer approach might be to use a Foxp3DTR transgenic mouse model, where Diphtheria toxin (DT) administration results in ablation of only Foxp3 expressing Treg cells. We are breeding these Foxp3DTR mice now for further tests in the future using CA170. Besides, we found that CA170 significantly increases CD8+ T cell infiltration. In the revision, we tested CA170 in a lung tumor model in which CD8 T cells had been depleted to evaluate the contribution of CD8 T cells on the effects of CA170 on lung tumorigenesis. This inhibitory effect of CA170 was decreased in mice in which CD8+ T cells had been depleted (Figure 2C), as evidenced by the increase in the lung tumor burden (Figure 2C). We have added these new data to the revision (Figure 2C).

Reviewers' Comments:

Reviewer #1:

Remarks to the Author:

I read the revised version of the manuscript, that I appreciate in the current form.

Only two suggested corrections, again on the difference between IC s and ICIs:

-page 4 line 64: immune checkpoints

-page 5 line 84: Immune checkpoint blockade

Reviewer #2:

None